# Patterns of diabetes mellitus by age, sex, and province among Iranian Hajj pilgrims and health care delivery during 2012–2022: A retrospective study of 469,581 participants

**Pirhossein Kolivand[1‡], Peyman Saberian[2‡], Hossein Saffari[3‡], Taher Doroudi[4,5], Ali Marashi[4], Masoud Behzadifar[6], Fereshteh Karimi[7], Soheila Rajaei[7], Behzad Raei[8], Seyed Jafar Ehsanzadeh[9], Arash Parvari[10], Samad Azari[11,7] ***

1 Department of Health Economics, Faculty of Medicine, Shahed University, Tehran, Iran, 2 Department of Anesthesiology, Imam Khomeini Hospital Complex, Tehran University of Medical Sciences, Tehran, Iran, 3 Hasheminejad Kidney Center (HKC), Iran University of Medical Sciences (IUMS), Tehran, Iran, 4 Iranian Red Crescent Society, Haj Medical Center, Tehran, Iran, 5 Shefa Neuroscience Research Center, Khatam Al anbia Hospital, Tehran, Iran, 6 Social Determinants of Health Research Center, Lorestan University of Medical Sciences, Khorramabad, Iran, 7 Research Center for Emergency and Disaster Resilience, Red Crescent Society of the Islamic Republic of Iran, Tehran, Iran, 8 Razi Educational and Therapeutic Center, Tabriz University of Medical Science, Tabriz, Iran, 9 English Language Department, School of Health Management and Information Sciences, Iran University of Medical Sciences, Tehran, Iran, 10 Department of Epidemiology and Biostatistics, School of Public Health, Tehran University of Medical Sciences, Tehran, Iran, 11 Hospital Management Research Center, Health Management Research Institute, Iran University of Medical Sciences, Tehran, Iran

‡ PK, PS and HS Contributed equally as the first authors
* Samadazari1010@gmail.com

## Abstract

### Objective

Hajj is among the oldest pilgrimages in the world, there is a limited study that evaluates the epidemiological pattern of Diabetes Mellitus [1] and the medical care required and provided to pilgrims. The present study assessed the prevalence and pattern of DM in Iranian pilgrims from 2012–22.

### Method

All demographic information, risk factors, and the prevalence of DM were extracted from the database and medical records of the Hajj Pilgrimage Medical Centre, Iranian Red Crescent Society through file reading. Also, to investigate the effect of the risk factors considered in the study, the multiple logistic regression model was used.

### Results

The present study included data from 469,581 Hajj pilgrims. Most pilgrims were in the age group of 45 to 70 years (73.25%). The prevalence of diabetes in patients over 70 years old was the highest (16.73%). The prevalence of DM was estimated at 14.64% in women and 12.51% in men. The lowest DM prevalence was in Lorestan (7.81%), North Khorasan (9.07%), Sistan and Baluchistan (9.29%), and Hamedan (9.41), respectively. The highest

**Data Availability Statement:** All relevant data are within the paper and its Supporting Information file.

**Funding:** The author(s) received no specific funding for this work.

**Competing interests:** The authors have declared that no competing interests exist.

prevalence rate was in Khuzestan (20.12%), Yazd (19.14), and Mazandaran (17.55), respectively. Our analysis reveals that, for instance, with each yearly increase in the age of the pilgrims (assuming other variables remain constant), the odds of having DM increase by 0.04. For the gender, the odds of having DM among women is 0.33 higher than among men, when the other variable is constant.

## Conclusions

The study results show a significant difference in the prevalence of diabetes in age, gender, and distribution in different provinces. Therefore, appropriate screening, diagnosis, and management by primary care physicians are necessary to prevent adverse health outcomes and reduce the economic burden of mortality and morbidity.

## 1. Introduction

The Hajj, an annual pilgrimage, is one of the largest mass gatherings worldwide, attracting over 2.5 million pilgrims from 200 countries to perform specific rituals in Mecca, the Kingdom of Saudi Arabia [2]. Successful participation in this ritual requires meticulous preparations undertaken months in advance. The increasing number of pilgrims challenges global and public health security for KSA and countries of various nationalities and races. These challenges encompass housing, food, water, transportation, communication, sanitation, mass-gathering and crowd control, and security, with the density of people reaching up to 7 individuals per square meter [3, 4].

Diabetes Mellitus [1] is a serious, chronic disease characterized by elevated blood glucose concentrations due to the effects of abnormal β-cell biology on insulin action. DM is one of the leading causes of death and disability worldwide, affecting individuals regardless of their country, age group, or sex. In 2017, the expenses incurred by health systems due to diabetes were estimated to be USD 727 billion. The number of people suffering from diabetes was estimated to be 285 million in 2009, 415 million in 2015, and 529 million (95% uncertainty interval [UI] 500–564) in 2021, indicating a continuous increase in global cases [5–7]. The prevalence of diabetes is high in several countries with a majority or large Muslim population, such as Pakistan, Indonesia, Egypt, Bangladesh, Iran, and India. All these countries rank among the top ten in the world in terms of diabetes prevalence. Based on a global data analysis, it is estimated that 148 million Muslims are affected by diabetes [8, 9].

Mass-gathering medicine at Hajj is challenged by issues of high morbidity, healthcare accessibility, patient management, and evacuation, especially in emergencies [10, 11]. Limited studies have evaluated the epidemiological pattern of diabetes and the medical care required by, and provided to pilgrims. The present study evaluated the prevalence and pattern of DM in Iranian Hajj pilgrims during the years 2012–22. Given the high population of pilgrims in the study, the results can be used to develop preventive and health protocols. This study will provide a comprehensive overview of diabetes status in Iranian pilgrims over ten years.

## 2. Methodology

### 2.1 Ethics approval and consent to participate

This study is a component from the research project with the Code of Ethics IR.RCS. REC.1401.023 from the Iranian Red Crescent Society, which has been conducted at the

Research Center for Emergency and Disaster Resilience, Red Crescent Society of the Islamic Republic of Iran, Tehran, Iran. Due to the retrospective nature of the study claims, data were analyzed anonymously, and written informed consent was waived according to the Research Ethics Committee of the Iranian Red Crescent Society before the start of the study. All stages of the study, including the design, implementation, and reporting, were conducted without any involvement from patients or the public. All methods were carried out according to Helsinki guidelines and regulations.

## 2.2 Research method

The present study was conducted in 2023. In this study, the prevalence of DM was examined by gender, province, and year from 2012 to 2022. The Islamic Republic of Iran's Red Crescent is responsible for conducting thorough medical examinations of Iranian Hajj pilgrims prior to their departure to Saudi Arabia. This screening follows a specific protocol established by the Medical Board of the Red Crescent, which includes clinical and para-clinical tests to create health profiles for each pilgrim. These profiles assess for various health conditions such as cardiac diseases, hypertension, respiratory diseases, diabetes, and psychiatric disorders. Additionally, influenza and pneumococcal vaccinations are administered to all pilgrims 15 days before their trip.

The Red Crescent also implements a syndromic surveillance system to monitor the health of pilgrims throughout their journey. This system tracks the health status of pilgrims, focusing on 19 specific diseases. When pilgrims experience health issues during the pilgrimage, they are first examined by physicians in their travel caravans, with referrals to Iranian hospitals in Mecca and Medina if necessary. Physicians in the caravans can perform essential lab tests as needed. The health data collected during these processes is crucial for identifying disease trends and planning future health measures for Hajj pilgrims.

All demographic information, risk factors, and the prevalence of diabetes were extracted from the databases of the Hajj and Pilgrimage Medical Center through file reading. Overall, this study investigated the age distribution of pilgrims, the prevalence of DM by age and gender among pilgrims in each age group, and the prevalence of DM among pilgrims by province and year.

## 2.3 Sampling and inclusion & exclusion criteria

No sampling was made in this study, and all Hajj pilgrims during the study period (from 2012 to 2022) were included in the analysis. We excluded Pilgrims whose information was not fully recorded in the databases. Data were accessed for research purposes on 25.04.2023. It should be noted that after the Mina incident in 2015, Iranian pilgrims did not participate in the 2016 Hajj pilgrimage. Due to the COVID-19 pandemic, no foreign pilgrims participated in the 2020 and 2021 pilgrimages. There is hence no data for those years.

## 2.4 Age category

In the present study, pilgrims were categorized into four age groups, under 15 years, 15–44 years, 45–69 years, and over 70-year-old pilgrims, respectively.

## 2.5 Data analysis and statistical tests

The extracted data was analyzed using Stata 17. The figure was generated by R software version 4.3.1 with the raster, geodata, rgeos, ggplot2, dplyr, and readxl packages. All data were cleaned and checked for accuracy. Initial descriptive analyses were performed to summarize the data

using frequency distributions. To investigate the effect of risk factors such as fasting blood sugar (FBS), age, and sex on the odds of diabetes among pilgrims, a multiple logistic regression model was applied. Since the diabetes status of individuals is a dichotomous variable (0 and 1), logistic regression was deemed an appropriate model for this study. Independent variables included in the model were selected based on clinical relevance and prior research findings. Considering that the prevalence of DM among pilgrims varies across provinces, and pilgrims within the same province are more similar to each other, the province was treated as a clustering variable. Robust standard errors were used to adjust for the correlation between pilgrims within the same province, accounting for intra-cluster correlation. All statistical analyses were conducted with a predetermined significance level of 5% ($\alpha = 0.05$).

## 3. Results

### 3.1 Characteristics of the study sample

In total, the present study included data from 469,581 hajj pilgrims, about 50% females and 50% males. Of the total pilgrims, 20.92% were hypertensive, and 13.55% had DM. The highest number of pilgrims, 89,492 individuals, belonged to the year 2019, and 73.25% were in the group aged 45–70 years. The age average was 55.48 ± 11.24 years (Table 1).

### 3.2 Distribution of DM by gender and age

According to the findings, the prevalence of DM among pilgrims increases with age, with the highest prevalence of the disease observed in pilgrims aged 70 years and above (Table 2). In terms of gender, 46% of diabetic patients were men and 54% were women.

**Table 1. Characteristics of Iranian Hajj pilgrims.**

| Variables | (%)N |
|---|---|
| **Gender** | |
| Male | 234,773 (49.99%) |
| Female | 234,808 (50.01%) |
| | **Age** (years) |
| <15 | 133 (0.02%) |
| 15–44 | 75,063 (15.98%) |
| 45–69 | 343,977 (73.25%) |
| 70 $\geq$ | 50,408 (10.75%) |
| Mean ± SD | 55.48 ± 11.24 |
| | **Diabetes** |
| Yes | 63,637 (13.55%) |
| No | 405,944 (86.45%) |
| | **Hypertension** |
| Yes | 98,264 (20.92%) |
| No | 371,317 (79.08%) |
| | **Year** |
| 2012 | 61,456 (13.09%) |
| 2013 | 61,364 (13.07%) |
| 2014 | 61,853 (13.17%) |
| 2016 | 71,053 (15.13%) |
| 2017 | 85,416 (18.19%) |
| 2019 | 89,492 (19.06%) |
| 2022 | 38,947 (8.29%) |

**Table 2. Distribution of DM in hajj pilgrims by age and gender.**

| Age (years) | DM, N (%) | | Total |
|---|---|---|---|
| | No | Yes | - |
| <15 | 133 | 0(0.0%) | 133 |
| 15≤age<45 | 72,761 | 2,211(2.94%) | 74,972 |
| 45≤age<70 | 290,497 | 53,006(15.43%) | 343,503 |
| 70 ≥ | 41,919 | 8,420(16.72%) | 50,339 |

### 3.3 Distribution of DM by province

As indicated by the data in the graph, the prevalence of diabetes among Hajj pilgrims from Lorestan province was the lowest among the 31 provinces. Of the 7110 pilgrims who journeyed to Mecca between 2012 and 2022, only 555 were diagnosed with diabetes, resulting in an estimated disease prevalence of 7.81%. Conversely, Khuzestan province exhibited the highest prevalence of this disease. Out of 24290 pilgrims, 4888 were affected by diabetes, yielding a prevalence rate of 20.12% in this province (Table 3). These findings highlight the significant regional disparities in diabetes prevalence among Hajj pilgrims (Fig 1).

### 3.4 The insulin consumption pattern

Among the 38,947 pilgrims in 2022, 5,198 had diabetes. Among this number of diabetic patients, 7 types of insulin were injected. The highest type was related to Insulin Glargine (44.1%) and Insulin Aspart(41.3%) and the lowest Insulin Detemir (0.1%). Table 4 shows the results of this section

### 3.5 DM healthcare delivery and economic burden

DM has been one of the most important risk factors and diseases among Hajj pilgrims. DM can raise morbidity and mortality and lead to an increase in the consumption of medical services and the economic burden of disease (Direct cost and indirect cost).

The services provided to diabetic patients (including general and specialized visits, medicine and prescription, paraclinical services, nursing, relief and transportation, etc.) to Hajj pilgrims in 2017, 2018, and 2022 were reported as 3207, 3087, and 2727, respectively (Fig 2).

### 3.6 Factors associated with DM disease

The impact of demographic variables and certain risk factors on DM has been evaluated. Given that the presence of diabetes is a dichotomous variable (DM = 0 for absence, DM = 1 for presence), a logistic regression model was employed. The data exhibited a clustered structure, suggesting that individuals within a province may share similarities in terms of certain environmental and genetic characteristics. As the objective of the study was not to draw inferences about the provinces, robust standard errors (Robust SE) were utilized to adjust for this clustering, thereby facilitating more accurate inferences (Table 5). This approach ensures a rigorous analysis while accounting for the inherent structure of the data.

The chi-square statistic for the overall model was estimated to be 7623.526, indicating that the fitted model is statistically significant (p-value<0.001). The null hypothesis, which posits that all independent variables in the model do not affect the odds (probability) of having DM, has been rejected. This suggests that at least one of the independent variables in the model significantly influences the odds (probability) of developing DM. The p-value for all independent

**Table 3. Distribution of DM by province among Hajj pilgrims.**

| Province | DM | | Total |
|---|---|---|---|
| | **0** | **1** | |
| Chahar Mahaal and Bakhtiari | 2,179 | 314 | 2,493 |
| | (87.40) | (12.60) | (100) |
| Kohgiluyeh and Boyer-Ahmad | 738 | 119 | 857 |
| | (86.11) | (13.89) | (100) |
| Alborz | 6,854 | 1,135 | 7,989 |
| | (85.79) | (14.21) | (100) |
| Ardabil | 3,473 | 584 | 4,057 |
| | (85.61) | (14.39) | (100) |
| Bushehr | 4,661 | 733 | 5,394 |
| | (86.41) | (13.59) | (100) |
| East Azarbayjan | 13,264 | 1,982 | 15,246 |
| | (87.00) | (13.00) | (100) |
| Fars | 19,277 | 2,875 | 22,152 |
| | (87.02) | (12.98) | (100) |
| Gilan | 3,139 | 609 | 3,748 |
| | (83.75) | (16.25) | (100) |
| Golestan | 18,937 | 2,787 | 21,724 |
| | (87.17) | (12.83) | (100) |
| Hamadan | 11,807 | 1,227 | 13,034 |
| | (90.59) | (9. 41) | (100) |
| Hormozgan | 4,938 | 656 | 5,594 |
| | (88.27) | (11.73) | (100) |
| Ilam | 617 | 94 | 711 |
| | (86.78) | (13.22) | (100) |
| Esfahan | 38,020 | 6000 | 44,020 |
| | (86.37) | (13.63) | (100) |
| Kerman | 14,930 | 2,604 | 17,534 |
| | (85.15) | (14.85) | (100) |
| Kermanshah | 4,779 | 671 | 5,450 |
| | (87.69) | (12.31) | (100) |
| Khuzestan | 19,402 | 4,888 | 24,290 |
| | (79.88) | (20.12) | (100) |
| Kurdistan | 7,906 | 1,093 | 8,999 |
| | (87.75) | (12.15) | (100) |
| Lorestan | 6,555 | 555 | 7,110 |
| | (92.19) | (7.81) | (100) |
| Markazi | 8,159 | 1,080 | 9,239 |
| | (99.86) | (0.14) | (100) |
| Mazandaran | 17,510 | 3,727 | 21,237 |
| | (82.45) | (17.55) | (100) |
| North Khorasan | 6,534 | 652 | 7,186 |
| | (90.93) | (9.07) | (100) |
| Qazvin | 5,448 | 960 | 6,408 |
| | (85.02) | (14.98) | (100) |
| Qom | 11,710 | 1,649 | 13,359 |
| | (87.66) | (12.34) | (100) |

(*Continued*)

**Table 3.** (Continued)

| Province | DM | | Total |
|---|---|---|---|
| | **0** | **1** | |
| **Razavi Khorasan** | 49,566 | 8,020 | 57,586 |
| | (87.07) | (13.93) | (100) |
| **Semnan** | 5,707 | 866 | 6,573 |
| | (86.82) | (13.18) | (100) |
| **Sistan and Baluchistan** | 8,399 | 860 | 9,259 |
| | (90.71) | (9.29) | (100) |
| **South Khorasan** | 8,953 | 1,300 | 10,253 |
| | )87.32( | (12.68) | (100) |
| **Tehran** | 63,383 | 8,655 | 72,038 |
| | (87.99) | (12.01) | (100) |
| **West Azarbayjan** | 17,713 | 2,908 | 20,621 |
| | (85.90) | (14.10) | (100) |
| **Yazd** | 13,371 | 3,166 | 16,537 |
| | (80.86) | (19.14) | (100) |
| **Zanjan** | 7,381 | 868 | 8,249 |
| | (89.48) | (10.52) | (100) |
| **Total** | 405,310 | 63,637 | 468,947 |
| | (86.43) | (13.57) | (100) |

variables in the model was less than 0.05, implying that all variables significantly affect the odds of developing DM.

Our analysis reveals that, with each yearly increase in the age of the pilgrims (assuming other variables remain constant), the odds of having DM increase by 0.04. For the gender, the odds of having DM among women is 0.33 higher than among men, when the other variables are constant. Similarly, for the fasting blood sugar variable, the odds of having DM for individuals with a fasting blood sugar level greater or equal to 126 (FBS≥126) are 29.84 times the odds of having DM for individuals with a fasting blood sugar level less than 126 (FBS<126), when other variables are considered to be constant.

## 4. Discussion

This study aimed to determine the prevalence of diabetes in Iranian Hajj pilgrims. This study showed that 13.55% of pilgrims had diabetes mellitus [1]. The prevalence of diabetes increases with age and reaches its peak in people over 70 years old. In terms of gender, 46% of diabetic patients were male and 54% were female. Lorestan province had the lowest prevalence with 7.81% and Khuzestan province had the highest prevalence with 20.12%. These findings emphasize significant regional disparities in the prevalence of diabetes among Iranian Hajj pilgrims.

Therefore, the findings of the present study showed that the effect of age on the prevalence of diabetes was significant. Other studies have investigated the effect of age on the prevalence of diabetes in Iran. A study conducted among individuals aged 35 to 70 years, using data from the Persian cohort study, estimated the age-standardized prevalence of diabetes at 15%, which aligns with the findings of the current research [12]. A similar study on the trend of DM prevalence among individuals aged 25 to 64 years reported the prevalence of diabetes as 8.4%, 9%, 11.1%, and 13.2% in 2004, 2007, 2011, and 2016, respectively [13]. The discrepancy between

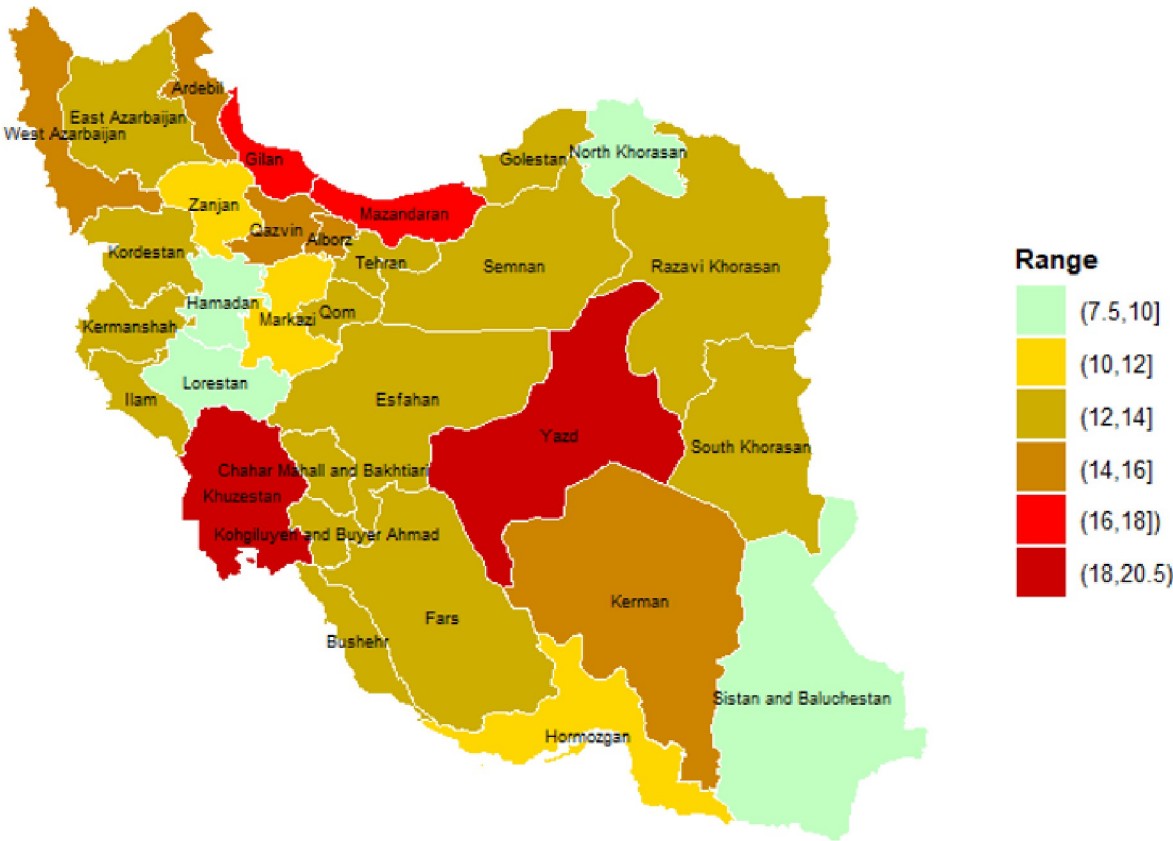

**Fig 1. Prevalence (%) of DM by province.** This figure depicts the prevalence percentage of Diabetes Mellitus across different provinces, showcasing regional variations in DM rates.

these findings and the current analysis could be attributed to the age differences of the study groups.

A study conducted in Kerman province revealed that the overall prevalence of DM is 12%, ranging from 7.1% in the age group of 15–24 years to 18.4% in the age group of 55–64 years [14]. The first phase of this study was carried out from 2009 to 2011 on 5,900 adults aged 15–75 in Kerman, and the second phase was conducted 5 years later, from 2014 to 2018, on a larger sample of 9,959 adults aged 15-80.The prevalence of DM in phase one was estimated at 10.8% and in phase two at 10.3% [14].

In 2015, it was estimated that there were 415 million people with diabetes between the ages of 20 to 79 years, 5 million deaths due to diabetes, and three-quarters of those with diabetes

**Table 4. Dose and number of insulin injections.**

| Dosage | Drug Name | (%) |
|---|---|---|
| 100 IU/ml | Insulin (Regular) | (4.9) |
| 100 IU/ml | Insulin Aspart | (41.3) |
| 100 IU/ml | Insulin Biphasic Isophane | (5.6) |
| 100 IU/ml | Insulin Detemir | (0.1) |
| 300 IU/ml | Insulin Glargine | (44.1) |
| 100 IU/ml | Insulin Glulisin | (1.6) |
| 100 IU/ml | Insulin Isophane | (1.6) |
| **Total** | | **100** |

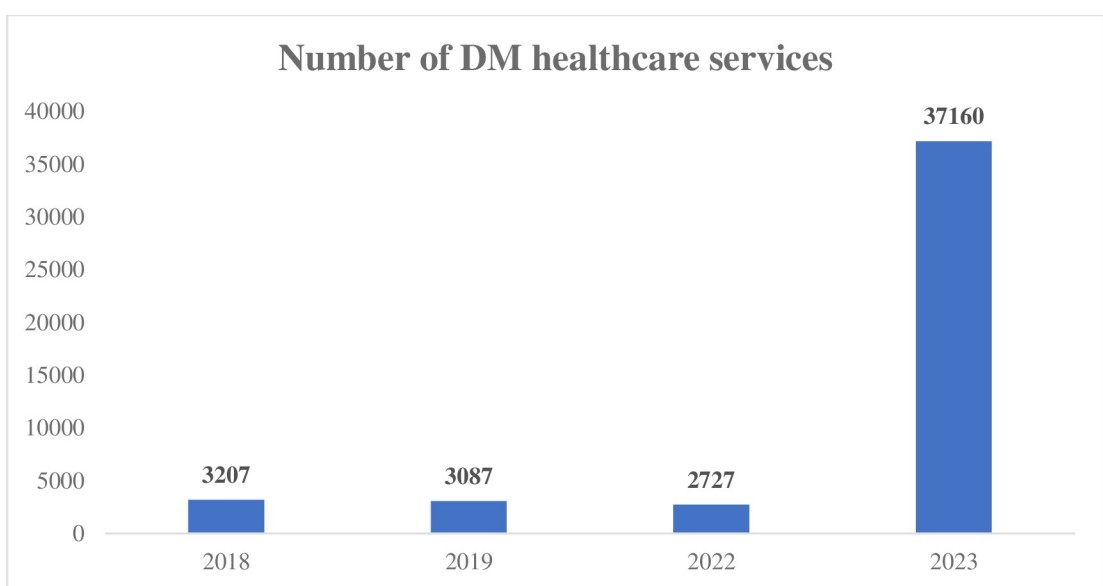

**Fig 2. Trend of DM healthcare delivery.** This figure showcases the changing trends in healthcare delivery for diabetes mellitus, emphasizing shifts in treatment methods, hospitalization rates, and patient outcomes over time.

lived in low- and middle-income countries. It is predicted that the number of people with diabetes between the ages of 20 to 79 will reach 642 million by the year 2040 [15].

The second finding of this study reveals that the prevalence of diabetes among women is higher than among men, with 12.51% of male pilgrims and 14.61% of female pilgrims diagnosed with this disease. A similar national-level study aligns with these findings, reporting the prevalence of DM as 13.5% in men, 14.7% in women, and 14.2% in both genders [16]. A systematic review, which reported the prevalence of DM in 25 papers, found rates ranging from 1.6% among Omani pilgrims to 40% among Chinese pilgrims. The weighted cumulative prevalence of DM among all pilgrims was estimated at 5% and hypertension was at 12%. This review study reported the lowest rate of DM prevalence among Australian and Iranian pilgrims [17].

Another study assessing the health status of Iranian pilgrims before departing for Hajj over different years showed an increasing trend of diabetes prevalence from 2.44% in 2004 to 8.31%

**Table 5. Results of logistic regression analysis for factors influencing the prevalence of DM.**

| DM | OR | Robust. St. Err. | t-value | p-value | [95% Conf | Interval] | Sig |
|---|---|---|---|---|---|---|---|
| Age | 1.039 | .003 | 13.83 | 0 | 1.034 | 1.045 | *** |
| Sex (Women = 1) | 1.329 | .041 | 9.34 | 0 | 1.252 | 1.411 | *** |
| FBS ($\geq$126) | 29.842 | 1.407 | 72.02 | 0 | 27.208 | 32.732 | *** |
| Constant | .005 | .001 | -37.26 | 0 | .004 | .006 | *** |
| Mean dependent var | | 0.136 | SD dependent var | | | 0.342 | |
| Pseudo r-squared | | 0.303 | Number of obs | | | 468947 | |
| Chi-square | | 7623.526 | Prob > chi2 | | | 0.000 | |
| Akaike crit. (AIC) | | 259495.904 | Bayesian crit. | | | 259540.137 | |

*** p < .01

** p < .05

* p<

in 2008 [18]. A significant study on French pilgrims revealed that the rate of diabetes in French pilgrims is five times higher than that in the general population, with the prevalence of diabetes, hypertension, and obesity being higher in female pilgrims than in male pilgrims [19]. The prevalence of chronic diseases is higher in Arab and Muslim countries than in non-Muslim countries. Furthermore, a large portion of the participants in the Hajj pilgrimageare over 60 years old [20]. A study on chronic diseases of deceased hospitalized individuals in Saudi hospitals during the Hajj pilgrimage showed that one-third of the deceased individuals suffered from diabetes [21].

A study in Malaysia showed that 26.5% of Hajj pilgrims in that country suffer from hypertension and 15.4% from diabetes [22]. The findings of a review of 22 selected studies show that the main health problems of pilgrims during Hajj are pneumonia, influenza, and asthma, and diseases such as stroke and heart attack (16.67%) and diabetes (13.33%) [23]. The study by Khan and colleagues shows that analgesics, antibiotics, antacids, antihistamines, as well as diabetes and blood pressure medications, were the most commonly prescribed drugs for Indian pilgrims [10]. Evidence from a longitudinal study on South African pilgrims indicates that high blood pressure and mostly type 2 diabetes are the most common diseases in these individuals, with prevalence rates of 22.6% and 13.2% respectively [24]. A study on the risk factors for hospitalization and mortality among Indonesian pilgrims shows that diabetes significantly increases the risk of hospitalization in Hajj pilgrims [25].

The third finding of this study pertains to the variations in diabetes prevalence among pilgrims from different provinces. A study indicates that the lowest prevalence of diabetes is in Kermanshah province, while the highest rate is observed in Khuzestan province. This aligns with the findings of the current study, which also reports the highest rate of diabetes prevalence among pilgrims from Khuzestan province [16]. However, a recently published study presents different findings, showing that the highest and lowest prevalence of diabetes are respectively in Yazd and Urmia. Another study reports that the highest prevalence of prediabetes was observed in Sari, while the lowest was in the city of Fasa [12]. These findings underscore the significant regional disparities in diabetes prevalence among Hajj pilgrims.

The fourth finding of this study pertains to the risk factors and determinants of DM prevalence. The results of the current study indicate that a higher probability of developing DM is associated with increasing age, high fasting blood sugar in pilgrims, and female gender. A study in this field reveals that the prevalence of DM is higher in urban residents compared to rural ones and in individuals with lower education levels [16]. According to the findings of an internal study, age and gender are determinants of the 5-year incidence rate of diabetes, such that increasing age, high blood pressure, overweight, and obesity are associated with an increased risk of diabetes [26].

The results of multivariate logistic regression in the study by Johari and colleagues in Fars province [27] showed that the prevalence of DM is linked with increasing age, being single, family history of diabetes, abdominal obesity, and hypertension. An inverse statistical relationship was observed between the prevalence of DM and factors such as high triglycerides, living in rural areas, higher education, and employment [28]. The significant increase in diabetes has occurred in all countries and both rural and urban areas. Accurate global, regional, and local estimates and predictions of DM prevalence are essential for planning and monitoring prevention and treatment strategies and assessing progress toward achieving the goals set globally for the control of non-communicable diseases and sustainable development.

In a prospective observational study, 26% of individuals hospitalized in the hospitals of the city of Mecca were diagnosed with diabetes, while previously they were unaware of their diabetes. Upon admission of patients, it was found that approximately 77% of the diabetic participants had poor diabetes control and about 55% had received diabetes education sessions

before going to Hajj. The study findings showed that approximately 37% of hospitalized individuals sometimes monitor their blood sugar themselves and only 22% knew that self-monitoring of blood sugar during the Hajj pilgrimageis recommended [29]. These findings underscore the significant impact of demographic and health factors on the prevalence of DM among Hajj pilgrims.

Another finding of the study was related to the investigation of the dose and units of insulin consumption. Insulin should be used within a maximum of 28 days and its temperature should be between 4˚C and 24˚C. If insulin is not used, it should not be frozen or placed near the freezer. Also, for maximum effectiveness, it should be kept in a refrigerator at a temperature of about 4˚C. Patients should pay attention to the dose insulin prescribed in the country of origin, as other countries may use different measurement tools. Many countries use a dose of 40 U/ml, but in Saudi Arabia, a dose of 100 U/ml is more common. Prior to travel, pilgrims should check with their health care provider for dosage requirements [30].

Some studies examined the challenges faced by diabetic pilgrims in ensuring the proper transportation and storage of insulin with the aim of maintaining its potency. Diabetic pilgrims should acquire adequate knowledge on how to store and store insulin during the travels. It is recommended that cool packs be used to maintain insulin at optimal temperature during transportation and storage. Pilgrims should also be adequately educated on insulin management strategies to minimize non-maintenance complications and ensure its quality [30, 31].

Hajj, one of the five pillars of Islam, is a mandatory religious duty for Muslims that must be performed at least once in their lifetime. Considering the global prevalence of diabetes at 8.8%, and the number of adult Muslims performing Hajj (approximately 2.5 million), it can be estimated that the number of Muslims with diabetes performing Hajj may exceed 220,000 annually. Islamic rulings stipulate that Hajj should not impose undue hardship on Muslims. The Holy Quran specifically exempts those who are physically or financially incapable from performing Hajj if it would result in harmful consequences for the individual. This exemption could apply to individuals with diabetes, given the severe and chronic complications associated with the condition. During Hajj, adherence to dietary guidelines, fluid intake, and mobility is crucial. Poor dietary control in pilgrims with diabetes can lead to serious complications and potentially fatal outcomes [9].

## 5. Conclusion

Accurate global, regional, and national estimates of diabetes prevalence are crucial for the planning and monitoring of prevention and treatment strategies, as well as for evaluating progress toward the goals set by joint global action plans for non-communicable disease control and sustainable development. Based on the studies, the present study is the most comprehensive to date, examining the prevalence of diabetes in Iranian pilgrims over the past decade, encompassing about half a million Iranian pilgrims. The results of the study reveal a significant variation in the prevalence of diabetes across different age groups, genders, and provinces. However, there appears to be a dearth of in-depth information on lifestyle adaptations and behavioral determinants among Hajj pilgrims concerning the management of chronic diseases during the Hajj. Therefore, appropriate screening, diagnosis, and management by primary care physicians are essential to prevent adverse health outcomes and mitigate the economic burden.

## Supporting information

**S1 Data.**
(XLSX)

## Acknowledgments

We appreciate the Hajj Pilgrimage Medical Centre, Iranian Red Crescent Society for offering the data used in our study. We also thank the respected reviewers who improved the quality of the article with their valuable comments. The data analyzed in this study can be accessed by sending a request to the corresponding author and the Hajj Pilgrimage Medical Centre, Iranian Red Crescent Society.

## Author Contributions

**Conceptualization:** Pirhossein Kolivand, Hossein Saffari, Ali Marashi, Soheila Rajaei, Samad Azari.

**Data curation:** Peyman Saberian, Taher Doroudi, Fereshteh Karimi, Soheila Rajaei, Arash Parvari, Samad Azari.

**Formal analysis:** Arash Parvari.

**Investigation:** Pirhossein Kolivand, Ali Marashi.

**Methodology:** Hossein Saffari, Masoud Behzadifar.

**Software:** Arash Parvari.

**Supervision:** Pirhossein Kolivand.

**Validation:** Peyman Saberian, Ali Marashi, Arash Parvari, Samad Azari.

**Visualization:** Masoud Behzadifar.

**Writing – original draft:** Pirhossein Kolivand, Peyman Saberian, Hossein Saffari, Masoud Behzadifar, Fereshteh Karimi, Soheila Rajaei, Behzad Raei, Samad Azari.

**Writing – review & editing:** Taher Doroudi, Fereshteh Karimi, Behzad Raei, Seyed Jafar Ehsanzadeh, Arash Parvari, Samad Azari.

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
