## [Decision Letter · Decision Letter 0]

23 Apr 2024

PONE-D-24-05546Patterns of Diabetes Mellitus by Age, Sex, and Province among Iranian Hajj Pilgrims and Health Care Delivery during 2012-2022: A Retrospective Study of 469,581 ParticipantsPLOS ONE

Dear Dr. Azari,

Thank you for submitting your manuscript to PLOS ONE. After careful consideration, we feel that it has merit but does not fully meet PLOS ONE’s publication criteria as it currently stands. Therefore, we invite you to submit a revised version of the manuscript that addresses the points raised during the review process.

We look forward to receiving your revised manuscript.

Kind regards,

Harunor Rashid, MD

Academic Editor

PLOS ONE

Journal Requirements:

3. We note that Figure 1 in your submission contain map images which may be copyrighted. All PLOS content is published under the Creative Commons Attribution License (CC BY 4.0), which means that the manuscript, images, and Supporting Information files will be freely available online, and any third party is permitted to access, download, copy, distribute, and use these materials in any way, even commercially, with proper attribution. For these reasons, we cannot publish previously copyrighted maps or satellite images created using proprietary data, such as Google software (Google Maps, Street View, and Earth). For more information, see our copyright guidelines: http://journals.plos.org/plosone/s/licenses-and-copyright.

We require you to either present written permission from the copyright holder to publish these figures specifically under the CC BY 4.0 license, or remove the figures from your submission:

Reviewers' comments:

Reviewer's Responses to Questions

**Comments to the Author**

1. Is the manuscript technically sound, and do the data support the conclusions?

Reviewer #1: No

Reviewer #2: Yes

2. Has the statistical analysis been performed appropriately and rigorously? 

Reviewer #1: No

Reviewer #2: Yes

3. Have the authors made all data underlying the findings in their manuscript fully available?

Reviewer #1: Yes

Reviewer #2: Yes

4. Is the manuscript presented in an intelligible fashion and written in standard English?

Reviewer #1: No

Reviewer #2: Yes

5. Review Comments to the Author

**Reviewer #1:** Major revisions are required. contradictions in results. the study lacks depth in methodology, and lacks many details. Should be meticulously revised before submitting. In this stage the manuscript is not suitable for acceptance and will affect the reputation of journal.

**Reviewer #2:** There is good data in this study. Some comments:

1. Indicate why 2020/2021 data is not available (we know its due to COVID-19)

2. Is there any indication of %age pilgrims who used insulin?

3. If insulin usage is known was the pathology/morbidity increased in that group?

4. Any indication of knowledge of insulin storage/handling during travel? (there was a recent study by Yezli on it).

6. PLOS authors have the option to publish the peer review history of their article (what does this mean?). If published, this will include your full peer review and any attached files.

Reviewer #1: No

Reviewer #2: No

---

## [Author Response · Author response to Decision Letter 0]

13 Jul 2024

Dear Editor,

The authors of the article have carefully checked the comments of the Reviewers and made the necessary changes in the article. These changes have been highlighted in the text. Additionally, the responses have been presented in the below table. We would like to express our gratitude to the referees for their valuable contributions, which have significantly enhanced the quality of the article.

Respectfully,

[Dr Samad Azari]

Table 1- The Journal Requirements

Num Comment Response

1 Please ensure that your manuscript meets PLOS ONE's style requirements Thanks for your reminder, all the text was reviewed according to the journal guidelines and the necessary corrections were made.

2 Please provide a complete Data Availability Statement in the submission form Data Availability Statement was added. 

3 We note that Figure 1 in your submission contain map images which may be copyrighted. All PLOS content is published under the Creative Commons Attribution License (CC BY 4.0), which means that the manuscript, images, and Supporting Information files will be freely available online, and any third party is permitted to access, download, copy, distribute, and use these materials in any way, even commercially, with proper attribution. For these reasons, we cannot publish previously copyrighted maps or satellite images created using proprietary data, such as Google software (Google Maps, Street View, and Earth). For more information, see our copyright guidelines: http://journals.plos.org/plosone/s/licenses-and-copyright It should be mentioned that in order to draw the figure in this manuscript, we used R software version 4.3.1 and the following packages under their respective licenses: `raster` (GPL-3), `geodata` (GPL-3), ` rgeos` (GPL-2), `ggplot2` (MIT), `dplyr` (MIT), and `readxl` (MIT). We acknowledge and thank the developers of these packages for their contributions to the open-source community.

We must emphasize that the presented map was not copied from any other source and there was no need to obtain permission from any original source. Therefore, we adhere to the Creative Commons Attribution 4.0 International (CC BY) license and presenting the map of this research has no conflict with the CCBY license and according to journal rules, the manuscript, images, and Supporting Information files will be freely available online, and any third party is permitted to access, download, copy, distribute, and use these materials in any way, even commercially, with proper attribution.

Table 2- The Responses to The Reviewer’s Comments

Num Comment Response

1 Reviewer #1: 

1. Major revisions are required. 

2. contradictions in results. 

3. the study lacks depth in methodology, and lacks many details. 

4. Should be meticulously revised before submitting. In this stage the manuscript is not suitable for acceptance and will affect the reputation of journal. Thanks to the precious comments of the respected reviewer. 

This descriptive and analytical study examines the information related to all Iranian Hajj pilgrims over ten years with more than half a million people's data. This survey includes many Variables Such as Age, Gender, Province, Diabetes Status, Blood Pressure, Insulin Use, and healthcare delivery. Therefore, in addition to the long study period that increases the reliability of the results, this study has also examined various important variables. These results can be used by the country's health system policymakers and also can be used by interested organizations such as the Red Crescent Society or the Ministry of Health in changing and improving screening and intervention policies related to Hajj. However, according to the valuable opinions of the reviewers, changes were made to the study method at this stage. The data analysis section was rewritten. Additionally, more details were provided in the study results section. In this part, the results related to the use of different types of insulin and their doses were also presented. Accordingly, the discussion section was also modified. 

Thank you again for your valuable comments. The research team welcomes your expert feedback. Undoubtedly, your comments will improve the quality of this research.

2 Reviewer #2: There is good data in this study. Some comments:

1. Indicate why 2020/2021 data is not available (we know its due to COVID-19)

2. Is there any indication of %age pilgrims who used insulin?

3. If insulin usage is known was the pathology/morbidity increased in that group?

4. Any indication of knowledge of insulin storage/handling during travel? (There was a recent study by Yezli on it). Thanks for the valuable comments of the respected reviewer.

As the respected reviewer has correctly pointed out, the data of some years were excluded in this study. This information was related to the Mina accident in 2015 and the information related to period of the Covid-19 pandemic. According to the honourable comment of the reviewer, relevant explanations were added in the methodology section. 

Regarding the reviewer's comment about insulin use, the desired section was added to the study findings section. In this section, the research findings on the units and types of insulin used among pilgrims are presented. Also, the valuable topics about insulin mentioned by the referee were examined in the discussion section of the study.

Thank you again for your valuable comments. These comments led to the improvement of research quality.

---

## [Decision Letter · Decision Letter 1]

20 Aug 2024

PONE-D-24-05546R1Patterns of Diabetes Mellitus by Age, Sex, and Province among Iranian Hajj Pilgrims and Health Care Delivery during 2012-2022: A Retrospective Study of 469,581 ParticipantsPLOS ONE

Dear Dr. Azari,

Thank you for submitting your manuscript to PLOS ONE. After careful consideration, we feel that it has merit but does not fully meet PLOS ONE’s publication criteria as it currently stands. Therefore, we invite you to submit a revised version of the manuscript that addresses the points raised during the review process.

Thank you for revising the manuscript. Reviewers' comments are more favorable now, but in places there are some factual and syntactical errors, once addressed those we would be happy to reconsider the manuscript for publication.

We look forward to receiving your revised manuscript.

Kind regards,

Harunor Rashid, MD

Academic Editor

PLOS ONE

Journal Requirements:

Reviewers' comments:

Reviewer's Responses to Questions

**Comments to the Author**

1. If the authors have adequately addressed your comments raised in a previous round of review and you feel that this manuscript is now acceptable for publication, you may indicate that here to bypass the “Comments to the Author” section, enter your conflict of interest statement in the “Confidential to Editor” section, and submit your "Accept" recommendation.

Reviewer #2: (No Response)

Reviewer #3: All comments have been addressed

2. Is the manuscript technically sound, and do the data support the conclusions?

Reviewer #2: Yes

Reviewer #3: Yes

3. Has the statistical analysis been performed appropriately and rigorously? 

Reviewer #2: Yes

Reviewer #3: Yes

4. Have the authors made all data underlying the findings in their manuscript fully available?

Reviewer #2: Yes

Reviewer #3: Yes

5. Is the manuscript presented in an intelligible fashion and written in standard English?

Reviewer #2: Yes

Reviewer #3: Yes

6. Review Comments to the Author

Reviewer #2: The minor comments in the initial draft were NOT addressed. I have put it in word format in the attached document.

Some of these include:

1. The word 'ceremony' should be deleted.

2. India is NOT a Muslim majority country.

3. Hajj WAS held in 2015, 2020 and 2021. Please correct that.

4. Iranian pilgrims DID attend 2015 Hajj. I met a number of them.

5. Iranian pilgrims did not attend the 2016 Hajj.

6. The paragraph below is not clear. Your study was till 2022, yet you include 2023. Does the below paragraph refer only to Iranians and only dialysis and why the massive difference for 2023?

Reviewer #3: Overall, the is article is so unique and has an important outcome that worth to be published. Minor modification could enrich the results and strengthens the findings of the study.

- Introduction:

o Hajj is attracting about 2.5 million Hajj pilgrims. NOT 3.5.

- Methodology:

o More information about the database is required:

Were the data was available for all Iranian pilgrims?

What are the medical data?

There was a mention about FBS in the results. More information is required about such data and how and when was collected?

What other information in database related to health records available can be added to this analysis?"

- Results:

o Were there any data about the compliance of diabetic pilgrims to treatment?

o When mentioning the rate of receiving DM services, provide the following:

percentage out of the total provided services NOT numbers each year.

percentage of dietetic pilgrims required services out of all dietetic pilgrims.

Detailed information about the medical services attained by diabetic patients would be very helpful? Incidence of specific complications, Acquiring extra medications, modification of treatment?"

- Table 5:

o Detailed information need to be added about the acquisition of FBS. Was it collected from all pilgrims? Was it collected once? and when that was? Is there any follow-up data?"

- Discussion:

o It is better to emphasize the main finding of the study briefly in the first paragraph, then discuss the findings in following paragraphs.

7. PLOS authors have the option to publish the peer review history of their article (what does this mean?). If published, this will include your full peer review and any attached files.

Reviewer #2: No

Reviewer #3: No

---

## [Author Response · Author response to Decision Letter 1]

14 Sep 2024

Thanks to the valuable comments of the respected reviewers, the answers to the comments are provided in this file. In the introduction section, corrections were made. In the method section, the necessary parts were added. In the findings section, the comments were answered. In the discussion section, the items desired by the reviewers were applied. Also, the details of some studies were added and reference 16 was removed.

Reviewer #2: The minor comments in the initial draft were NOT addressed. I have put it in Word format in the attached document.

Some of these include:

1. The word 'ceremony' should be deleted.

Answer: Thank you for your valuable comments. revised.

2. India is NOT a Muslim majority country.

Answer: T Thank you for your valuable comments. his item was checked. India is also one of the ten countries that has the most Hajj pilgrims. Related articles are provided below.

o The prevalence of diabetes is high in several nations with large Muslim population, such as Pakistan, Indonesia, Egypt, Bangladesh and India, and all of these countries rank amongst the top ten countries in the world in terms of diabetes prevalence

o Shaikh S, Ashraf H, Shaikh K, Iraqi H, Ndour Mbaye M, Kake A, Mohamed GA, Selim S, Wali Naseri M, Syed I, Said JA. Diabetes care during Hajj. Diabetes Therapy. 2020 Dec; 11:2829-44.

3. Hajj WAS held in 2015, 2020 and 2021. Please correct that.

Answer: Thank you for your valuable comments. This item was corrected. It should be noted that there was a mistake in converting the date.

4. Iranian pilgrims DID attend 2015 Hajj. I met a number of them.

Answer: Thank you for your valuable comments. This item was corrected. It should be noted that there was a mistake in converting the date.

5. Iranian pilgrims did not attend the 2016 Hajj.

Answer: Thank you for your valuable comments. This item was corrected. It should be noted that there was a mistake in converting the date.

6. The paragraph below is not clear. Your study was till 2022, yet you include 2023. Does the below paragraph refer only to Iranians and only dialysis and why the massive difference for 2023?

Answer: No, we mean all health services provided, including general and specialized visits, medicine and prescription, nursing, relief and transportation, etc., which are available and reported in 2018, 2019, 2022. The part related to 2023 was deleted.

Please clarify. Did 2727 Iranian pilgrims need dialysis in 2022 and 37160 need it 2023? Do these numbers refer to Iranian pilgrims only? Please rewrite.

Answer: It means all services that were provided to diabetic pilgrims. This part has been rewritten.

Reviewer #3: 

Introduction:

o Hajj is attracting about 2.5 million Hajj pilgrims. NOT 3.5.

Answer: This item was corrected in the article.

o Check currency. None of the suggestions in my first review have been incorporated. Was the PDF that I sent checked?

Answer: This item was corrected in the article.

o India is not a muslim amjority country

Answer: This item was checked. India is also one of the ten countries that has the most Hajj pilgrims. Related articles are provided below.

- The prevalence of diabetes is high in several nations with large Muslim population, such as Pakistan, Indonesia, Egypt, Bangladesh and India, and all of these countries rank amongst the top ten countries in the world in terms of diabetes prevalence

- Shaikh S, Ashraf H, Shaikh K, Iraqi H, Ndour Mbaye M, Kake A, Mohamed GA, Selim S, Wali Naseri M, Syed I, Said JA. Diabetes care during Hajj. Diabetes Therapy. 2020 Dec; 11:2829-44.

o Check currency. None of the suggestions in my first review have been incorporated. Was the PDF that I sent checked?

Answer: This item was corrected in the article.

- Methodology:

o More information about the database is required:

Were the data was available for all Iranian pilgrims?

What are the medical data?

There was a mention about FBS in the results. More information is required about such data and how and when was collected?

What other information in database related to health records available can be added to this analysis?"

Answer: This description was added to the method section» The Islamic Republic of Iran's Red Crescent is responsible for conducting thorough medical examinations of Iranian Hajj pilgrims prior to their departure to Saudi Arabia. This screening follows a specific protocol established by the Medical Board of the Red Crescent, which includes clinical and para-clinical tests to create health profiles for each pilgrim. These profiles assess for various health conditions such as cardiac diseases, hypertension, respiratory diseases, diabetes, and psychiatric disorders. Additionally, influenza and pneumococcal vaccinations are administered to all pilgrims 15 days before their trip. The Red Crescent also implements a syndromic surveillance system to monitor the health of pilgrims throughout their journey. This system tracks the health status of pilgrims, focusing on 19 specific diseases. When pilgrims experience health issues during the pilgrimage, they are first examined by physicians in their travel caravans, with referrals to Iranian hospitals in Mecca and Medina if necessary. Physicians in the caravans can perform essential lab tests as needed. The health data collected during these processes is crucial for identifying disease trends and planning future health measures for Hajj pilgrims«.

- Results:

o Were there any data about the compliance of diabetic pilgrims to treatment?

o When mentioning the rate of receiving DM services, provide the following:

percentage out of the total provided services NOT numbers each year.

percentage of dietetic pilgrims required services out of all dietetic pilgrims.

Detailed information about the medical services attained by diabetic patients would be very helpful? Incidence of specific complications, Acquiring extra medications, modification of treatment?"

Answer: Detailed information about the medical services obtained by diabetic patients, such as the special complications, receiving additional drugs, treatment modification, should be done by checking the patients' files case by case. Therefore, access to the desired information of the respected reviewer was not possible in this study. We hope that future studies will investigate the desired aspects. Also, because the information about services was not available for all years, we could not provide the percentage of services provided in each year in this section.

- Table 5:

o Detailed information needs to be added about the acquisition of FBS. Was it collected from all pilgrims? Was it collected once? and when that was? Is there any follow-up data?"

Answer: Screening of pilgrims before travel includes clinical and paraclinical tests to create a health profile for each pilgrim. In this screening, various health conditions are evaluated. When pilgrims experience health problems during the Hajj journey, they are first examined by physicians in their traveling caravans and, if necessary, refer to Iranian hospitals in Mecca and Medina. Doctors in the caravans can perform necessary laboratory tests if needed. Information about treatment complexity, treatment compliance, and other information for each individual is recorded in their file. It is possible to access this information of patients in the form of file reading and case by case. Therefore, access to the desired information of the respected reviewer was not possible in this study. We hope that future studies will investigate the desired aspects.

Fig 3: 

o Should range not be written as 7.5-10 then 10-12?

- Answer: This item was corrected in the article.

- Discussion:

o It is better to emphasize the main finding of the study briefly in the first paragraph, then discuss the findings in following paragraphs.

- Answer: This item was corrected in the article. Also, the items desired by the reviewers were applied. Also, the details of some studies were added and reference 16 was removed.

---

## [Editor Report · Decision Letter 2]

19 Sep 2024

Patterns of Diabetes Mellitus by Age, Sex, and Province among Iranian Hajj Pilgrims and Health Care Delivery during 2012-2022: A Retrospective Study of 469,581 Participants

PONE-D-24-05546R2

Dear Dr. Azari,

We’re pleased to inform you that your manuscript has been judged scientifically suitable for publication and will be formally accepted for publication once it meets all outstanding technical requirements.

Kind regards,

Harunor Rashid, MD

Academic Editor

PLOS ONE
---

## [Editor Report · Acceptance letter]

27 Sep 2024

PONE-D-24-05546R2 

PLOS ONE

Dear Dr. Azari, 

I'm pleased to inform you that your manuscript has been deemed suitable for publication in PLOS ONE. Congratulations! Your manuscript is now being handed over to our production team.

Kind regards, 

on behalf of

Dr. Harunor Rashid 

Academic Editor

PLOS ONE